# What are the determinants of variation in caretaker satisfaction with sick child consultations? A cross-sectional analysis in five low-income and middle-income countries

Anne-Marie Turcotte-Tremblay [1,2] Hwa-Young Lee [3,4] Margaret E Kruk [2]

¹Faculty of nursing, Université Laval, Quebec, Quebec, Canada
²Department of Global Health and Population, Harvard T. H. Chan School of Public Health, Boston, Massachusetts, USA
³Graduate School of Public Health and Healthcare Management, The Catholic University of Korea, Seoul, Republic of Korea
⁴Catholic Institute for Public Health and Healthcare Management, The Catholic University of Korea, Seoul, Republic of Korea

**Correspondence to**
Dr Hwa-Young Lee;
diana0224@gmail.com;
diana0224@catholic.ac.kr

## ABSTRACT

**Objectives** The objective of this study was to explore determinants of variation in overall caretaker satisfaction with curative care for sick children under the age of 5 in five low-income and middle-income countries.

**Design** A pooled cross-sectional analysis was conducted using data from the Service Provision Assessment.

**Setting** We used data collected in five countries (Afghanistan, Democratic Republic of the Congo, Haiti, Malawi and Tanzania) between 2013 and 2018.

**Participants** Respondents were 13 149 caretakers of children under the age of 5 who consulted for a sick child visit.

**Primary outcomes measured** The outcome variable was whether the child's caretaker was very satisfied versus more or less satisfied or not satisfied overall. Predictors pertained to child and caretaker characteristics, health system foundations and process of care (eg, care competence, user experience). Two-level logistic regression models were used to assess the extent to which these categories of variables explained variation in satisfaction. The main analyses used pooled data; country-level analyses were also performed.

**Results** Process of care, including user experience, explained the largest proportion of variance in caretaker satisfaction (13.8%), compared with child and caretaker characteristics (0.9%) and health system foundations (3.8%). The odds of being very satisfied were lower for caretakers who were not given adequate explanation (OR: 0.56, 95% CI 0.46 to 0.67), who had a problem with medication availability (OR: 0.31, 95% CI 0.27 to 0.35) or who encountered a problem with the cost of services (OR: 0.57, 95% CI 0.48 to 0.66). The final model explained only 21.8% of the total variance. Country-level analyses showed differences in variance explained and in associations with predictors.

**Conclusions** Better process of care, especially user experience, should be prioritised for its benefit regarding caretaker satisfaction. Unmeasured factors explained the majority of variation in caretaker satisfaction and should be explored in future studies.

## STRENGTHS AND LIMITATIONS OF THIS STUDY

⇒ This is the first multicountry analysis to examine the determinants of variation in caretaker satisfaction with healthcare services for sick children under the age of 5 in low-income and middle-income countries (LMICs).

⇒ The study has the advantage of combining multiple sources of data (facility audits, health provider interviews and exit interviews with caretakers).

⇒ Novel evidence shows that the process of care, including the technical quality of care and the caretaker experience, explains the largest proportion of variance in caretaker satisfaction with sick childcare visit across five LMICs.

⇒ The satisfaction rates reported by caretakers were generally high, possibly due to desirability bias in participants' responses during interviews, low patient expectations and behaviours during the audits.

## BACKGROUND

User satisfaction is an important dimension of high-quality health systems.[1] It reflects the ultimate judgement of the consumer and influences decisions on when and where to seek care.[1] Consequently, user satisfaction measures are increasingly used to evaluate the performance of healthcare systems, improve accountability and provide financial incentives to healthcare staff. The results of user satisfaction surveys are also used to help managers and healthcare providers identify service factors needing improvement.[2]

However, whether and to what extent user satisfaction measures are appropriate to evaluate health systems depends on the extent to which they are influenced by health system versus other factors. Studies have shown that user satisfaction is a complex and multidimensional concept with numerous determining factors. Batbaatar *et al*[2] reported that user satisfaction is influenced by technical

care, interpersonal care, physical environment, access, organisational characteristics, continuity of care and outcome of care. Focusing on child health, McCormick et al[3] identified the health status of the child, sociodemographic characteristics and history of treatment as predictors of parental satisfaction in the USA. However, these analyses have not paid much attention to broader factors (eg, facility characteristics, country-level characteristics) that may influence user satisfaction.

Understanding the relative contributions of different types of factors to explaining variation in user satisfaction is useful to better interpret and use results from user satisfaction surveys for quality improvement. However, few studies have examined how much variation is explained by determinants of user satisfaction in various countries. One study used data from 21 European Union countries to examine variation in user satisfaction with the health system.[4] Those authors found that all factors taken into account in their model, such as patient expectations, health status and personality, explained only 17.5% of the observed variation in user satisfaction, of which 10.4% was explained by user experience with system responsiveness.[4] They concluded that, contrary to published reports, user experience accounts for only a small fraction of variation in user satisfaction with the healthcare system. However, the proportion of variance in user satisfaction explained by health system and non-health system factors in low-income and middle-income countries (LMICs) remains unclear. Most evidence on this issue was derived in high-income countries.

The objectives of our study were twofold: (1) to assess the proportion of variance in caretaker overall satisfaction explained by factors related to child and caretaker characteristics, health system foundations, and process of care; (2) to identify variables associated with caretaker overall satisfaction in five LMICs.

## METHODS
### Study setting
Table 1 presents the demographic characteristics and healthcare situations in the five countries included in this study. All of these countries are characterised by high rates of neonatal mortality. Democratic Republic of the Congo (DRC) had the highest mortality rate for children under the age of 5 per 1000 live births (58) and the lowest health spending per capita (51 PPP$). In contrast, Malawi had the lowest mortality rate for children under the age of 5 (39) and the second lowest value for health spending per capita (82 PPP$).

### Data source and sampling
The data were obtained from Service Provision Assessment (SPA) surveys. SPA surveys are conducted by the Demographic and Health Survey Program of United States Agency for International Development, with a national statistics agency in the countries surveyed to measure the capacity of health systems in LMICs. The datasets are available online.[5–9] For the SPA survey, facilities are selected from a comprehensive list of health facilities in each country, categorised by facility type, managing authority and region; a nationally representative sample of health facilities is selected. A description of the methodology used for SPA surveys is available online.[10]

We limited our study population to the most recent SPA surveys that included a question on user satisfaction with consultation for sick children under the age of 5 in five countries: Afghanistan (2018), Democratic Republic of

| Table 1 Context of study countries | | | | | |
|---|---|---|---|---|---|
| | **Afghanistan** | **DRC** | **Haiti** | **Malawi** | **Tanzania** |
| Population (in millions)* | 38.0 | 86.8 | 11.3 | 18.6 | 58.0 |
| Urban population (% of total population)† | 26 | 46 | 57 | 17 | 35 |
| Gross domestic product per person (USD)* | 507 | 581 | 1273 | 412 | 1122 |
| Health expenditure per capita (PPP$)* | 286 | 51 | 143 | 82 | 99 |
| Primary education completion rate, total (% of relevant group)‡ | 84 | 70 | 49 | 80 | 69 |
| Literacy rate of adults (%) ‡§ | 43 | 77 | 62 | 62 | 78 |
| Birth attended by skilled health staff (%)‡ | 59 | 80 | 42 | 90 | 64 |
| Maternal mortality (per 100 000 live births)¶ | 638 | 473 | 480 | 349 | 524 |
| Neonatal mortality (per 1000 live births)* | 36 | 27 | 25 | 20 | 20 |
| Mortality rate under age 5 (per 1000 live births)† | 58 | 81 | 61 | 39 | 49 |
| Hospital beds (per 1000 people)‡ | 0.4 | 0.8 | 0.7 | 1.3 | 0.7 |

*Data from 2019.
†Data from 2020.
‡Data from the most recent year available.
§Percentage of people aged 15 and above able to read and write simple statements about everyday life.
¶Data from 2017.
DRC, Democratic Republic of the Congo.

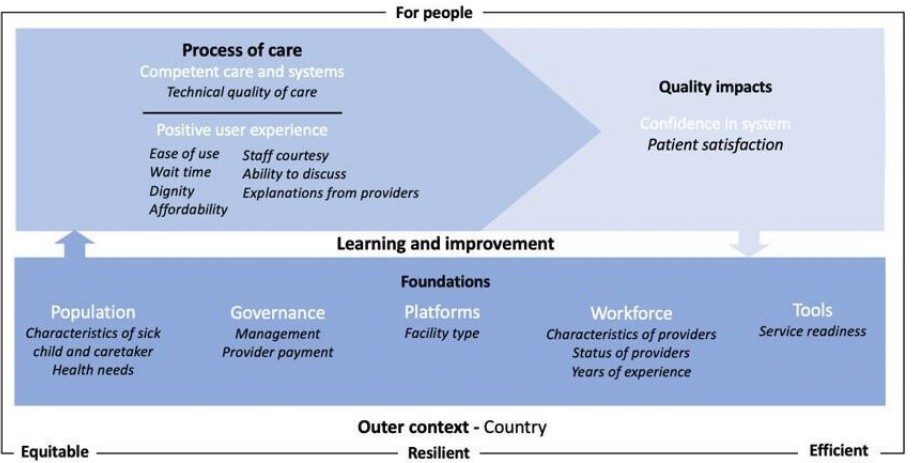

**Figure 1** Conceptual framework. Adapted from Kruk et al[1]

the Congo (DRC) (2018), Haiti (2017), Malawi (2013) and Tanzania (2015).

All caretakers who sought care for a sick child consultation in hospitals or primary healthcare facilities were included in our study without any specific exclusion criteria. We used data from four separate modules: facility audits, health provider interviews, observation of protocols for sick child care and exit interviews with caretakers. Facility audits collected data on available human resources, basic amenities and availability of essential equipment, supplies and medicines. A sample of health providers was selected from the facility roster to be interviewed and observed. Trained enumerators observed clinical visits to assess adherence to clinical guidelines during consultations for sick children. The caretakers (eg, parents, grandparents) of children observed during the consultations completed an exit interview regarding their level of satisfaction with the services received, problems encountered and demographic characteristics.

### Data management and quality control

The procedures for data management and quality were similar across countries.[11–15] The interviewers were supposed to review the data and enter it in tablets. The data files were transferred to a supervisor who oversaw the data collection process. When supervisors noted missing information or errors, they sent the data back to the interviewer for revision. Then, the data were sent to a central office via the Internet. In the central office, data processors detected inconsistencies and gave feedback to the team in the field to resolve the problems. For tracking of systematic errors arising from each interviewer, field check tables were run. If an interviewer committed errors systematically, the central office and coordinators followed up with the interviewers.[11–15]

### Framework

To conceptualise our analyses, we adapted the High Quality Health Systems framework from the Lancet Global Health Commission.[1] As shown in figure 1, this framework comprises three key domains: foundations, process of care and quality impacts.[1] Health system foundations include the population and their health needs, health sector governance, care delivery platforms, workforce skills and tools such as medicines and equipment. Process of care is composed of two subdomains: competent care and user experience. Competent care refers to systematic assessment, correct diagnosis, appropriate treatment, counselling, prevention and detection. Positive user experience refers to respect, autonomy, clear communication, short wait times, patient affordability and ease of use. Caretaker satisfaction, our outcome of interest, is a quality impact measure that contributes to confidence in the system and willingness to use healthcare.

### Outcome variable

The outcome variable is caretaker satisfaction, extracted from the exit interview module. The following question was used to elicit caretaker satisfaction regarding the sick child consultation: "In general, which of the following statements best describes your opinion of the services you either received or were provided at this facility today? Very satisfied, more or less satisfied, or not satisfied." Based on the low levels of dissatisfaction, we created a binary variable of satisfaction equal to 1 if the caretaker was very satisfied and 0 otherwise (ie, if the caretaker was more or less satisfied, or not satisfied).

### Predictor variables

We identified child and caretaker characteristics that were likely to influence user satisfaction by reviewing previous studies.[1–4] Child characteristics included gender (male or female), age (12 months or less, 13–59 months and age not specified), and severity of symptoms, constructed by summing the reasons the caretaker identified for bringing in the child, including cough, diarrhoea, fever, vomiting, problem with feeding, convulsions, excessive sleepiness or other symptoms (range: 1–8). Caretaker characteristics included age (≤19, 20–35, ≥36 years and age not specified) and education level (none, primary, secondary, tertiary or higher).

To characterise the facilities, we examined the types of management authorities (public, private not for profit, private for profit) and whether providers received a regular salary supplement. Facility types categorised into hospitals, health centres or clinics, health posts or dispensaries, and urban setting (vs rural) were included. To characterise the workforce, providers were separated into three categories: physicians/clinical officers/assistant medical officers; nurses/midwives; and aides/assistants. Other provider characteristics included gender and years since graduation (as a proxy for experience). To characterise the facilities' tools, we drew on data from facility audits to calculate a service readiness score for preventive and curative care regarding sick child health based on the WHO Service Availability and Readiness Assessment guidelines.[16] The overall score was calculated by averaging all items in four domains: staff and training, equipment, diagnostics, and medicines and commodities. The original scores ranged between 0 and 1 but were multiplied by 10 to facilitate interpretation (0–10). Online supplemental table 1 presents the full list of indicators used.

We assessed the technical quality of sick child consultations by investigating the degree to which the provider adhered to clinical guidelines during the consultation observation.[17 18] The technical quality score was calculated as the percentage of clinically recommended tasks performed in four domains: patient history taking, physical exam, drug administration and immunisation, and client education and counselling. We also multiplied the score by 10 to facilitate interpretation. The items included for technical quality indicators for sick child care are presented in the supplementary file (online supplemental tables 2,3).

Caretakers' perceptions of their experience during the visit were examined through exit interviews by asking whether they had encountered problems that day related to: (1) days or hours of service at this facility (ie, when it opened and closed); (2) wait time to see a provider; (3) facility cleanliness; (4) how staff treated them; (5) ability to discuss problems or concerns about the health issue; (6) amount of explanation received about the problem or treatment; (7) medication; and (8) cost of services or treatments. For each of these, responses were scored as the number of problems encountered. We included the country variable as a proxy for country-level outer context.

## Statistical analysis

We performed a complete case analysis, which yielded a total of 13 149 sick child visits across 3421 facilities for the sample in the 5 countries. For variables with the most missing data (eg, child age, caretaker age), we added the value 'not specified' to be able to include the cases in the analysis. Less than 5% of the sample had missing data, in total.

We calculated descriptive statistics for all variables of interest, presenting the proportions for categorical variables and mean and SD for continuous covariates. These descriptive analyses incorporated survey sampling weights for patients and facilities.

For the main analyses, we performed a two-level logistic regression analysis based on pooled data from the five countries, where level 1 was the consultation and level 2 was the facility. This provided variance estimates while accounting for clustering effect at the facility level. The following general structure was used:

$$logit\ (\Pr\ (Y_{ji} = 1|X) = \beta_0 + \beta \cdot X_{ij} + u_{0j}$$

where Y (caretaker satisfaction) and X (representing a vector of independent variables) were each assumed to follow a two-level data structure, with both fixed-effect ($\beta_0, \beta \cdot X_{ij}$) and random-effect parameters ($u_{0j}$). The random-effect parameter ($u_{0j}$) was assumed to follow a normal distribution, with mean of 0 and variances of $u_{0j} \sim N\ (0, \sigma^2_{u_0}$. We confirmed the linearity between the four continuous independent variables and log-odds of the outcome using the Box-Tidwell test.

Four models were estimated based on the general modelling structure outlined above, by incrementally adding blocks of covariates following our conceptual framework (see online supplemental table 4) in the supplementary file for details on the modelling strategy). M0 was a null/unadjusted model with an intercept term in the fixed part of the model. M1 added child and caretaker characteristics. M2 built on the previous model to include variables related to health system foundations. M3 added variables related to the process of care. For the analyses with the pooled data, M4 included the country as a variable.

We applied Snijders and Bosker's method to measure the explained proportion of variance in caretaker satisfaction for the logistic multilevel model (details are provided in online supplemental box of the online supplemental file).[19] Since logistic regression models do not have a level 1 residual term, unexplained variance at the consultation level was estimated as $\pi^2/3 \approx 3.29$ based on the method summarised by Goldstein *et al*.[8] When considering only the unexplained part of the total variance, the proportion attributed to the health facilities level (called residual intraclass correlation) was calculated.

We also performed country-stratified analyses to examine whether the results were country specific. All analyses were conducted in Stata V.16.0 (StataCorp).

## Patient and public involvement statement

Patients and/or the public were not involved in the design, or conduct, or reporting, or dissemination of this research.

## RESULTS

### Descriptive statistics

Table 2 presents summary statistics for the sample. Overall, 77% of caretakers were very satisfied. The percentage of very satisfied respondents was highest in Haiti (89%)

**Table 2** Descriptive statistics per country

| Variables | Afghanistan N (%) | DRC N (%) | Haiti N (%) | Malawi N (%) | Tanzania N (%) | Total N (%) |
|---|---|---|---|---|---|---|
| Very satisfied/total analytic sample, N (very satisfied, %) | 345/556 (62) | 1859/2592 (71) | 1850/2093 (89) | 2581/3177 (82) | 3538/4731 (75) | 10 171/13 149 (77) |
| **Child and caretaker characteristics** | | | | | | |
| Number of symptoms caretaker identified (0–8), Mean (SD) | 3.2 (1.3) | 3.4 (1.4) | 2.9 (1.4) | 2.5 (1.2) | 2.7 (1.3) | 2.8 (1.3) |
| Child is female | 236 (43%) | 1238 (47%) | 974 (47%) | 1566 (50%) | 2333 (49%) | 6346 (48%) |
| Age of child (months) | | | | | | |
| 12 or less | 263 (47%) | 959 (37%) | 1026 (49%) | 1300 (41%) | 1974 (42%) | 5520 (42%) |
| 13–59 months | 246 (44%) | 1618 (62%) | 1045 (50%) | 1833 (58%) | 2768 (58%) | 7508 (57%) |
| Not specified | 47 (8%) | 33 (1%) | 15 (1%) | 21 (1%) | 8 (0%) | 123 (1%) |
| Age of caretaker (years) | | | | | | |
| 20–35 | 366 (66%) | 1799 (69%) | 1489 (71%) | 2471 (78%) | 3505 (74%) | 9628 (73%) |
| ≥ 19 | 19 (3%) | 222 (8%) | 166 (8%) | 324 (10%) | 439 (9%) | 1167 (9%) |
| 36+ | 110 (20%) | 424 (16%) | 407 (20%) | 249 (8%) | 719 (15%) | 1906 (14%) |
| Not specified | 62 (11%) | 167 (6%) | 24 (1%) | 111 (3%) | 87 (2%) | 449 (3%) |
| Education of caretaker | | | | | | |
| None | 308 (55%) | 569 (22%) | 231 (11%) | 385 (12%) | 901 (19%) | 2392 (18%) |
| Primary | 53 (9%) | 828 (32%) | 583 (28%) | 2021 (64%) | 3072 (65%) | 6556 (50%) |
| Secondary | 121 (22%) | 1065 (41%) | 1105 (53%) | 683 (22%) | 693 (15%) | 3665 (28%) |
| Tertiary or higher | 74 (13%) | 149 (6%) | 167 (8%) | 66 (2%) | 83 (2%) | 538 (4%) |
| **Health system foundations** | | | | | | |
| Provider is female | 14 (3%) | 576 (14%) | 991 (60%) | 568 (29%) | 2288 (45%) | 4435 (34%) |
| Provider type | | | | | | |
| Doctor, advanced practice clinician, paramedical | 460 (100%) | 1225 (31%) | 1196 (72%) | 1631 (83%) | 3112 (61%) | 7622 (58%) |
| Nurse, midwife | 0 (0%) | 2689 (67%) | 463 (28%) | 324 (16%) | 1938 (38%) | 5413 (41%) |
| Others—pharm, lab, dental, non-clinical | 0 (0%) | 90 (2%) | 4 (0%) | 11 (1%) | 12 (0%) | 115 (1%) |
| Provider receives regular salary supplements | 282 (61%) | 579 (14%) | 168 (10%) | 746 (38%) | 1461 (29%) | 3234 (25%) |
| Service readiness score for sick child care (0–10), mean (SD) | 6.0 (1.3) | 6.5 (1.8) | 5.7 (1.6) | 6.0 (1.6) | 5.2 (1.6) | 5.8 (1.7) |
| Facility type | | | | | | |
| Hospitals | 353 (61%) | 520 (22%) | 558 (23%) | 591 (16%) | 288 (7%) | 2308 (18%) |
| Health centres | 227 (39%) | 1861 (78%) | 1283 (53%) | 2804 (78%) | 627 (15%) | 6800 (52%) |
| Health posts and dispensaries | 0 (0%) | 0 (0%) | 563 (23%) | 195 (5%) | 3285 (78%) | 4042 (31%) |
| Facility management | | | | | | |
| Public | 157 (27%) | 1506 (63%) | 800 (33%) | 2334 (65%) | 3486 (83%) | 8281 (63%) |
| Private not-for-profit | 71 (12%) | 576 (24%) | 906 (38%) | 748 (21%) | 383 (9%) | 2682 (20%) |
| Private for-profit | 352 (61%) | 299 (13%) | 697 (29%) | 509 (14%) | 332 (8%) | 2187 (17%) |
| Urban facility | 575 (99%) | 446 (19%) | 1128 (47%) | 749 (21%) | 1070 (25%) | 3966 (30%) |
| **Process of care** | | | | | | |
| Technical quality of care observed (0–10), mean (SD) | 2.4 (1.1) | 3.4 (1.2) | 3.1 (1.2) | 3.0 (1.3) | 3.1 (1.5) | 3.1 (1.4) |
| Problem with days or hours services are provided | 73 (13%) | 326 (12%) | 128 (6%) | 601 (19%) | 612 (13%) | 1738 (13%) |

Continued

**Table 2** Continued

| Variables | Afghanistan N (%) | DRC N (%) | Haiti N (%) | Malawi N (%) | Tanzania N (%) | Total N (%) |
|---|---|---|---|---|---|---|
| Problem with wait time | 137 (25%) | 530 (20%) | 474 (23%) | 998 (32%) | 1329 (28%) | 3466 (26%) |
| Problem with cleanliness of facility | 92 (16%) | 481 (18%) | 100 (5%) | 250 (8%) | 769 (16%) | 1690 (13%) |
| Problem with how staff treated patient | 100 (18%) | 269 (10%) | 48 (2%) | 229 (7%) | 317 (7%) | 961 (7%) |
| Problem with ability to discuss concerns with provider | 77 (14%) | 323 (12%) | 180 (9%) | 354 (11%) | 397 (8%) | 1328 (10%) |
| Problem with amount of explanation provided | 75 (13%) | 372 (14%) | 163 (8%) | 356 (11%) | 401 (8%) | 1365 (10%) |
| Problem with medication availability | 156 (28%) | 581 (22%) | 205 (10%) | 635 (20%) | 1540 (32%) | 3115 (24%) |
| Problem with cost of services | 193 (35%) | 575 (22%) | 105 (5%) | 174 (6%) | 462 (10%) | 1508 (11%) |

DRC, Democratic Republic of the Congo.

and lowest in Afghanistan (62%). Approximately 68% of caretakers had primary education level or less. Of the providers, 34% were female, and 58% were qualified as doctors or advance practice clinicians. The majority of facilities were health centres (52%) and public (63%). The mean scores for technical quality of care observed were quite low (3.1). Because data in Afghanistan only cover urban areas, the percentage of highly qualified providers, hospitals and private for-profit facilities was notably higher than in the other countries (table 2).

## Total variance explained

Figure 2 shows the percentage of variance explained in each model. Variance explained in the fully adjusted model with the combined data from five countries was 21.8%. However, it varied significantly across countries, ranging from 9.0% in DRC to 51.7% in Afghanistan. Factors related to patient and caretaker characteristics only explained 0.9% of the total variance (0.8% in Tanzania~4.3% in Afghanistan). The explained variance increased by 3.8% when variables related to health system foundations, such as facility and provider characteristics, were included (0.9% in Malawi~10.8% Afghanistan). The largest percentage of the total variance was explained by process of care in both the pooled and country-stratified analyses. When process of care variables were added, the explained variance increased by 13.8% in the pooled analyses. Variance explained by process of care was notably higher in Afghanistan and Tanzania (36.6% and 24.2%, respectively) than in the other countries (5.8% in DRC~10.4% in Malawi).

## Determinants of caretaker satisfaction

Figure 3 shows the results of the final model's fixed part based on the pooled data from the five countries. More severe symptoms of the child (OR: 0.94, 95% CI 0.91 to 0.98) and higher education of caretakers (OR: 0.78 and 0.53, 95% CI 0.66 to 0.93 and 0.40 to 0.69 for secondary

and tertiary level or higher, respectively) were associated with lower odds of being very satisfied with the service received. Caretakers who sought care in hospital were less likely to be very satisfied with service received than those at lower level facilities. In particular, caretakers who visited health posts and dispensaries for their sick child were 1.82 times more likely to be very satisfied with the service than those who visited hospitals (OR: 1.33; 95% CI 1.11 to 1.59 and OR: 1.82; 95% CI 1.40 to 2.38, respectively). Caretakers whose child received care in facilities with higher service readiness scores were more likely to be very satisfied.

Problems with process of care were strong predictors of user satisfaction. Higher technical quality scores for sick child care increased the likelihood of being very satisfied (OR: 1.08, 95% CI 1.04 to 1.13 per 10% increase). Negative user experiences significantly decreased the odds of caretaker satisfaction. In particular, caretakers who perceived they were not given adequate explanation and perceived a problem with medication availability were 0.56 and 0.31 times less likely to be very satisfied with the service for their sick child, respectively (95% CI 0.46 to 0.67 and 0.27 to 0.35, respectively). Problems with the cost of services significantly reduced the odds of caretaker satisfaction (OR: 0.57, 95% CI 0.48 to 0.66). None of the providers' characteristics were associated with caretaker satisfaction. All country dummy variables, which served as proxies for national context, were significant. Compared with caretakers in Malawi, those in Afghanistan, DRC and Tanzania were less likely to be very satisfied with services, while those in Haiti were 1.76 times more likely to be very satisfied.

The results of the fixed country effects are presented in table 3. Tertiary or higher education was associated with satisfaction in Tanzania (OR: 0.4, 95% CI 0.2 to 0.8), Afghanistan (OR: 0.3, 95% CI 0.1 to 1.0) and DRC

 Turcotte-Tremblay A-M, *et al. BMJ Open* 2023;**13**:e071037. doi:10.1136/bmjopen-2022-071037

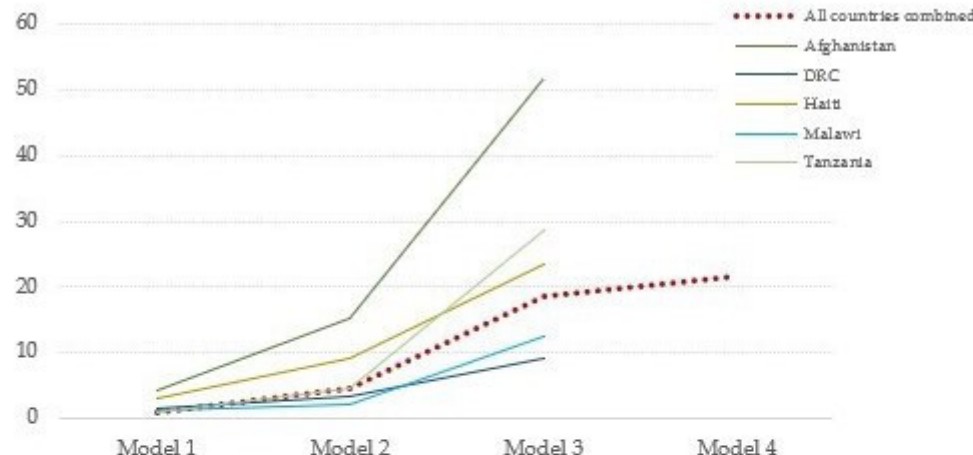

| | All countries combined | Afghanistan | DRC | Haiti | Malawi | Tanzania |
|---|---|---|---|---|---|---|
| Null model | 0.0 | 0.0 | 0.0 | 0.0 | 0.0 | 0.0 |
| Model 1 | 0.9 | 4.3 | 1.5 | 3.1 | 1.3 | 0.8 |
| Model 2 | 4.7 | 15.1 | 3.2 | 9.2 | 2.2 | 4.4 |
| Model 3 | 18.5 | 51.7 | 9.0 | 23.6 | 12.6 | 28.6 |
| Model 4 | 21.8 | NA | NA | NA | NA | NA |

**Figure 2** Variance explained with addition of variables to the model. DRC, Democratic Republic of the Congo. NA, not applicable

(OR: 0.2, 95% CI 0.1 to 0.7). In DCR, some facility characteristics appeared to be important determinants of satisfaction. More specifically, caregivers in health centres or private for-profit facilities had higher odds of being very satisfied compared with those seeking care in hospitals or public facilities (OR: 2.2, 95% CI 1.3 to 3.8; OR: 2.8, 95% CI 1.2 to 6.2, respectively). On the other hand, in DRC some aspects of the patients' experiences, such as wait time (OR: 0.5, 95% CI 0.3 to 1.0) and how staff treated patients (OR: 0.8, 95% CI 0.8 0.3 to 2.1) were not associated with caretaker satisfaction. Caregivers who had problems with how the staff treated them had lower odds of being very satisfied in Afghanistan, Haiti and Malawi. In all five countries, caregivers who encountered problems with medication availability and cost of services were significantly less likely to be very satisfied (see table 3).

### DISCUSSION

Based on combined data from five countries, we found that factors related to the process of care during consultations for children under the age of 5 explained the highest proportion of variance in caretaker satisfaction (13.8 %), followed by factors related to the health systems' foundations (3.8%). Patient and caretakers' characteristics

explained the smallest proportion of variance in caretaker satisfaction (0.9%). The children's number of symptoms and the caregiver's higher education levels both decreased the odds of being very satisfaction. The facility type and the readiness to provide sick child services also predicted overall satisfaction, although the providers' characteristics did not. In terms of process of care, the odds of being very satisfied were lower for caretakers who were not given adequate explanations and those who had encountered problems with medication availability and the cost of services.

To our knowledge, this is the first multicountry study to examine factors affecting variation in caretaker satisfaction with sick child consultations in multiple LMICs. Several findings deserve attention. First and foremost, in both the pooled and country-stratified analyses, user-reported problems with process of care accounted for the largest proportion of variance explained in caretaker satisfaction with sick child care. This finding is consistent with two previous studies. In Peru, Leslie *et al*[20] found that user experience accounted for a larger proportion of variance in all user satisfaction measures, including three and five categories of net promoter score and satisfaction level, than the proportion explained by individual, facility and contextual factors combined. Similarly, Bleich *et al*'s

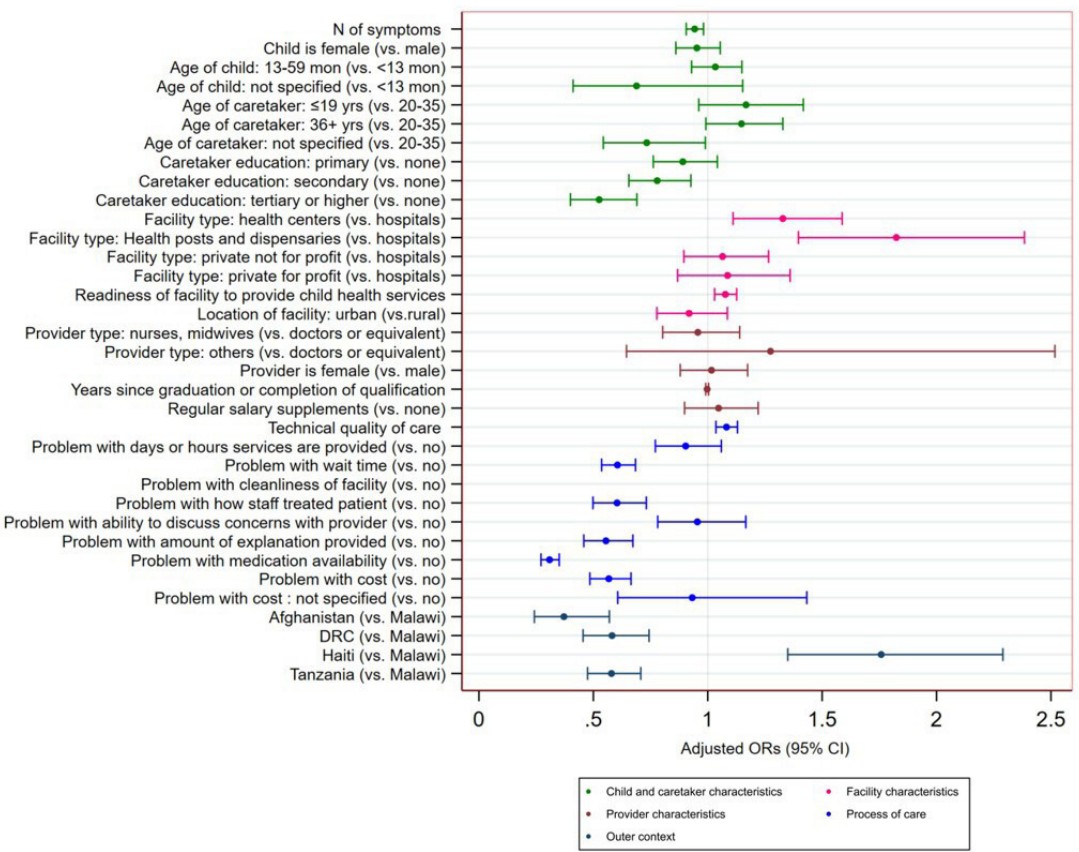

**Figure 3** Determinants of caretaker satisfaction. DRC, Democratic Republic of the Congo.

study[4] based on 21 European Union countries reported that all variables considered in the analyses (eg, user experience, user expectations) accounted for 17.5% of the observed variation in user satisfaction, of which 10% was explained by user experience.

Second, technical quality of care, a subdimension of process of care, was a significant predictor of caretaker satisfaction with the service. Odds of caretakers' being very satisfied with the service increased 1.08 times with every 10% increase in technical quality score—a substantial effect size if quality were 30% or 40% higher, which is conceivable given low baseline performance. However, this association was not significant in three of the five countries. The reason for this may be that, in some contexts, patients find it difficult to assess technical quality of care due to lack of medical expertise.[21] In contrast, it may be easier for patients to perceive tangible aspects, such as outcome quality (eg, how much the symptoms have been improved),[22 23] and assess their user experience (eg, wait times, affordability of care).[24–26] This should be taken into account when user satisfaction is used to assess the performance of facilities.

Third, we found substantial between-country heterogeneity. The proportion of the total variance explained by the variables was notably higher in Afghanistan (51.7 %, as opposed to 9.0%~28.6% in the other four countries). Also, the proportion of variance explained by process of care was highest in Afghanistan (increase in R squared:

36.6% in Afghanistan vs 5.8%–24.2%). Compared with other countries, the provider's courtesy and communication (eg, explanations, discussions) appeared to be particularly important aspects of caretaker satisfaction in Afghanistan. In DRC, total variance explained by the variables was only 9%, suggesting there were still more variables influencing caretaker satisfaction, which we were unable to take into account (eg, insecurity, unpaid government employees). According to Bleich et al,[4] external factors such as media coverage could influence how satisfied people are with their healthcare system. Future qualitative and mixed methods studies could be useful to explore and uncover new context-specific factors that affect variance in user satisfaction.

Factors predicting patient satisfaction also differed between countries. Experiences of problems with medication availability or cost were the only indicators related to user experience that had a marked negative association with caretaker satisfaction across all five countries. On the other hand, inability to discuss concerns with providers and problems with how staff treated patients showed significant negative association with caretaker satisfaction in only a few countries, whereas it was identified as a strong predictor of user satisfaction in earlier studies conducted on different types of maternal care in other countries. Specifically, Dzomeku found that providers' unfriendliness and ineffective communication negatively affected maternal satisfaction in Ghana.[27] Similar

**Table 3** Final multilevel models for each country

| | Afghanistan | | | DRC | | | Haiti | | | Malawi | | | Tanzania | | |
|---|---|---|---|---|---|---|---|---|---|---|---|---|---|---|---|
| | OR | | 95% CI | OR | | 95% CI | OR | | 95% CI | OR | | 95% CI | OR | | 95% CI |
| **Child and caretaker characteristics** | | | | | | | | | | | | | | | |
| Number of symptoms caregiver identified | 1.0 | ns | 0.9 to 1.3 | 0.8 | *** | 0.7 to 1.0 | 1.0 | ns | 0.9 to 1.1 | 0.9 | ns | 0.8 to 1.0 | 1.0 | ns | 0.9 to 1.1 |
| Child is female (vs male) | 0.6 | ns | 0.3 to 1.1 | 0.9 | ns | 0.6 to 1.4 | 1.1 | ns | 0.8 to 1.5 | 0.9 | ns | 0.6 to 1.3 | 1.1 | ns | 0.8 to 1.4 |
| Age of child (vs 12 months or less) | | | | | | | | | | | | | | | |
| 13–59 months | 1.3 | ns | 0.6 to 2.5 | 1.1 | ns | 0.7 to1.8 | 0.8 | ns | 0.6 to 1.2 | 0.8 | ns | 0.6 to 1.1 | 1.1 | ns | 0.9 to 1.4 |
| Not specified | 0.3 | * | 0.1 to 0.9 | 0.3 | ns | 0.1 to 1.2 | 1.0 | | | 0.8 | ns | 0.1 to 4.5 | 1.1 | ns | 0.0 to 41.2 |
| Age of caretaker (vs 20–35 years old) | | | | | | | | | | | | | | | |
| ≥19 years old | 2.5 | ns | 0.3 to 21.2 | 1.4 | ns | 0.7 to 3.2 | 0.8 | ns | 0.4 to 1.5 | 1.1 | ns | 0.7 to 1.6 | 1.5 | ns | 1.0 to 2.2 |
| 36+ years old | 1.3 | ns | 0.7 to 2.5 | 1.3 | ns | 0.8 to 2.3 | 1.1 | ns | 0.7 to 1.7 | 1.4 | ns | 0.9 to 2.4 | 0.8 | ns | 0.6 to 1.2 |
| Not specified | 1.3 | ns | 0.5 to 3.9 | 0.3 | *** | 0.1 to1.0 | 0.5 | ns | 0.1 to 2.0 | 1.0 | ns | 0.5 to 2.3 | 1.0 | ns | 0.3 to 3.4 |
| Education level of caretaker (vs none) | | | | | | | | | | | | | | | |
| Primary | 0.7 | ns | 0.2 to 1.9 | 0.6 | ns | 0.3 to 1.2 | 1.0 | ns | 0.4 to 2.4 | 0.9 | ns | 0.6 to 1.4 | 0.9 | ns | 0.6 to 1.3 |
| Secondary | 0.6 | ns | 0.3 to 1.5 | 0.4 | *** | 0.2 to 0.9 | 0.8 | ns | 0.4 to 1.6 | 0.7 | ns | 0.4 to 1.3 | 0.7 | ns | 0.5 to 1.1 |
| Tertiary or higher | 0.3 | * | 0.1 to 1.0 | 0.2 | *** | 0.1 to 0.7 | 0.5 | ns | 0.2 to 1.0 | 0.9 | ns | 0.4 to 2.0 | 0.4 | * | 0.2 to 0.8 |
| **Health system foundations** | | | | | | | | | | | | | | | |
| *Facility characteristics* | | | | | | | | | | | | | | | |
| Types (vs hospitals) | | | | | | | | | | | | | | | |
| Health centres | 1.6 | ns | 0.6 to 4.8 | 2.2 | *** | 1.3 to 3.8 | 1.5 | ns | 0.9 to 2.4 | 0.9 | ns | 0.6 to 1.4 | 1.3 | ns | 0.9 to 1.7 |
| Health posts and dispensaries | – | | | | | | 1.8 | ns | 1.0 to 3.5 | 1.4 | ns | 0.6 to 3.3 | 1.7 | * | 1.1 to 2.6 |
| Facility management (vs public) | | | | | | | | | | | | | | | |
| Private not-for-profit | 1.9 | ns | 0.4 to 9.8 | 1.1 | ns | 0.7 to 1.9 | 1.2 | ns | 0.8 to 1.9 | 0.8 | ns | 0.5 to 1.2 | 1.0 | ns | 0.7 to 1.4 |
| Private for-profit | 0.6 | ns | 0.2 to 2.1 | 2.8 | *** | 1.2 to 6.2 | 1.2 | ns | 0.7 to 1.9 | 0.8 | ns | 0.5 to 1.4 | 1.3 | ns | 0.7 to 2.4 |
| Readiness of facility to provide child services | 1.7 | *** | 1.3 to 2.1 | 1.1 | ns | 1.0 to 1.3 | 1.0 | ns | 0.9 to 1.1 | 1.0 | ns | 0.9 to 1.1 | 1.1 | ns | 1.0 to 1.2 |
| Urban facility (vs rural) | 1.4 | ns | 0.6 to 3.6 | 0.8 | ns | 0.4 to 1.4 | 0.5 | *** | 0.3 to 0.7 | 0.8 | ns | 0.5 to1.3 | 0.9 | ns | 0.6 to 1.1 |
| *Provider characteristics* | | | | | | | | | | | | | | | |
| Types (vs doctor, advanced pract. clinician) | 1.0 | | | | | | | | | | | | | | |
| Nurse, midwife | – | | | 0.7 | ns | 0.4 to 1.2 | 0.6 | ns | 0.4 to 1.1 | 0.9 | ns | 0.5 to 1.3 | 1.1 | ns | 0.8 to 1.6 |
| Other – pharm, lab, dental, non-clinical | – | | | 0.7 | ns | 0.2 to 2.5 | – | | | – | | | 0.7 | ns | 0.4 to 1.3 |
| Provider is female (vs male) | 1.8 | ns | 0.2 to 20.9 | 1.0 | ns | 0.5 to 2.0 | 0.8 | ns | 0.5 to 1.2 | 1.2 | ns | 0.8 to 1.9 | 1.0 | ns | 0.8 to 1.3 |
| Years since graduation or qualification | 1.0 | ns | 0.9 to 1.1 | 1.0 | ns | 1.0 to 1.0 | 1.0 | ns | 1.0 to 1.0 | 1.0 | ns | 1.0 to 1.0 | 1.0 | ns | 1.0 to 1.0 |
| Regular salary supplements for provider (vs none) | 1.2 | ns | 0.5 to 3.0 | 0.7 | ns | 0.3 to 1.5 | 1.3 | ns | 0.7 to 2.3 | 1.0 | ns | 0.7 to 1.4 | 1.1 | ns | 0.8 to 1.4 |
| **Process of care** | | | | | | | | | | | | | | | |
| Technical quality of care observed | 0.8 | ns | 0.6 to 1.1 | 1.0 | ns | 0.8 to 1.1 | 1.2 | * | 1.0 to 1.4 | 1.0 | ns | 0.9 to 1.1 | 1.2 | *** | 1.1 to 1.3 |

Continued

**Table 3** Continued

| | Afghanistan | | | DRC | | | Haiti | | | Malawi | | | Tanzania | | |
|---|---|---|---|---|---|---|---|---|---|---|---|---|---|---|---|
| | OR | | 95% CI | OR | | 95% CI | OR | | 95% CI | OR | | 95% CI | OR | | 95% CI |
| Problem with days or hours services are provided | 1.0 | ns | 0.4 to 2.7 | 1.5 | ns | 0.7 to 3.3 | 0.6 | ns | 0.3 to 1.1 | 0.8 | ns | 0.5 to 1.2 | 1.0 | ns | 0.7 to 1.5 |
| Problem with wait time | 0.5 | ns | 0.2 to 1.1 | 0.5 | ns | 0.3 to 1.0 | 0.3 | *** | 0.2 to 0.4 | 0.7 | * | 0.5 to 1.0 | 0.6 | *** | 0.4 to 0.7 |
| Problem with cleanliness of facility | 0.7 | ns | 0.2 to 2.2 | 0.9 | ns | 0.5 to 1.9 | 0.5 | * | 0.2 to 1.0 | 0.7 | ns | 0.4 to 1.0 | 0.6 | ** | 0.4 to 0.8 |
| Problem with how staff treated patient | 0.2 | *** | 0.1 to 0.5 | 0.8 | ns | 0.3 to 2.1 | 0.2 | *** | 0.1 to 0.5 | 0.4 | *** | 0.2 to 0.6 | 0.6 | ns | 0.4 to 1.0 |
| Problem with ability to discuss concerns with provider | 0.3 | ** | 0.1 to 0.9 | 2.2 | ns | 0.7 to 6.5 | 1.6 | ns | 0.8 to 2.9 | 0.6 | ns | 0.3 to 1.2 | 0.6 | * | 0.4 to 1.0 |
| Problem with amount of explanation provided | 0.2 | *** | 0.1 to 0.5 | 0.7 | ns | 0.3 to 1.9 | 0.4 | ** | 0.2 to 0.8 | 0.6 | ns | 0.3 to 1.2 | 0.4 | *** | 0.2 to 0.7 |
| Problem with medication availability | 0.4 | ** | 0.2 to 0.8 | 0.2 | *** | 0.1 to 0.3 | 0.5 | ** | 0.3 to 0.8 | 0.5 | *** | 0.3 to 0.7 | 0.2 | *** | 0.1 to 0.2 |
| Problem with cost (vs none) | 0.2 | *** | 0.1 to 0.5 | 0.5 | *** | 0.3 to 0.8 | 0.5 | ** | 0.2 to 0.9 | 0.6 | * | 0.3 to 1.0 | 0.5 | *** | 0.3 to 0.8 |
| Problem with cost not specified (vs none) | – | | | 0.9 | ns | 0.3 to 2.9 | – | | | 1 | | 0.4 to 2.4 | 1.21 | | 0.2 to 9.2 |

*p ≤ 0.05; **p≤0.01; ***p≤0.001. ns, not significant.
Numbers were rounded to one decimal place for space.

results were found in Lebanon and Gambia, where it was reported that women were more satisfied with services when they had the providers' attention and adequate communication with them.[28 29] As such, factors linked to patient satisfaction may depend on context, which is related to various expectations, public sentiment, values or norms.[2 30] These findings can inform strategic priorities in each country aimed at enhancing patient satisfaction with sick child consultations and possibly patient retention.

This study has several limitations. First, the satisfaction rates reported by caretakers were generally high, possibly due to desirability bias in participants' responses during interviews, low patient expectations and behaviours during the audits. The observation of the providers' performance during the consultation potentially biased the patient—provider interaction. Second, some potential predictors of caretaker satisfaction, such as patients' and providers' personalities and past experiences with the healthcare system, were not available from the SPA surveys and thus were not included in our analysis. Finally, because this is a cross-sectional analysis, the associations identified cannot be interpreted causally. Despite this limitation, our study has the advantage of combining multiple sources of data (facility audits, health provider interviews, exit interviews with caretakers). It is also the first multicountry analysis to examine the determinants of variation in caretaker satisfaction for sick child care in LMICs.

The study is useful to develop a research agenda. First, in our analyses, there is still a substantial proportion of unexplained variation in caretakers' satisfaction. Future research should explore new determinants of variation in patient satisfaction throughout children's care pathways. Second, the heterogeneity between countries supports the need for more studies on how cultural differences affect patient satisfaction.

## CONCLUSION

This study showed that the process of care explained the largest proportion of variance in caretaker satisfaction during sick child consultations in five LMICs. Caretakers who encountered problems related to waiting time, how the staff treated them, the amount of explanation provided, medication availability and cost of services were less likely to be very satisfied. High-quality healthcare systems should be able to satisfy users, including caretakers of sick children. People who are satisfied may have more confidence in their healthcare system and be more likely to seek care. In this regard, it is important to understand which factors shape variation in user satisfaction in different settings. Interventions targeting patient experience may be effective to improve patient satisfaction, thereby positively influencing utilisation rates and patient retention over the course of care.

**Acknowledgements** The authors express their gratitude to all actors involved in collecting and reporting Service Provider Assessment data in the participating countries.

**Contributors** AMTT developed the idea, conducted the analyses and wrote the first draft of the manuscript. She is the responsible for the overall content as the guarantor. She accepts full responsibility for the work and/or the conduct of the study, had access to the data, and controlled the decision to publish. HYL validated the results and reviewed the manuscript. MK proposed the original idea, provided the database, oversaw the analyses and reviewed the manuscript.

**Funding** This work was supported by the Bill & Melinda Gates Foundation (grants INV-005254). The first author received a fellowship from the Fonds de recherche du Québec – Santé.

**Disclaimer** These funding sources had no role in the design of this study the analyses or interpretation of the data.

**Competing interests** None declared.

**Patient and public involvement** Patients and/or the public were not involved in the design, or conduct, or reporting, or dissemination plans of this research.

**Patient consent for publication** Not applicable.

**Ethics approval** The original survey implementers obtained ethical approvals for data collection; the Harvard University Human Research Protection Program approved this secondary analysis as exempt from human subjects review, exempted this study. Participants gave informed consent to participate in the study before taking part.

**Provenance and peer review** Not commissioned; externally peer reviewed.

**Data availability statement** Data are available in a public, open access repository. The Service provider data used for these analyses are publicly available through the Demographic and Health Survey (DHS) Program of United States Agency for International Development (USAID). Before you can download datasets, you must register as a DHS data user on this website: https://dhsprogram.com/data/new-user-registration.cfm. The data for each country can be accessed online.

**ORCID iDs**
Anne-Marie Turcotte-Tremblay http://orcid.org/0000-0002-6138-9908
Hwa-Young Lee http://orcid.org/0000-0003-2591-1436
Margaret E Kruk http://orcid.org/0000-0002-9549-8432

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
