## [Reviewer comments · BMJ Open]

ARTICLE DETAILS

TITLE (PROVISIONAL)	What are the determinants of variation in caretaker satisfaction with sick child consultations? a cross-sectional analysis in five low- and middle-income countries
AUTHORS	Turcotte-Tremblay, Anne-Marie; Lee, Hwa-Young; Kruk, Margaret

VERSION 1 – REVIEW

REVIEWER	Tawfiq, Essa The University of Auckland, Department of Epidemiology and Biostatistics, School of Population Health
REVIEW RETURNED	05-Feb-2023

GENERAL COMMENTS	Overall Thank you for providing me the opportunity to review this paper. This study used secondary data from national representative samples from five LMIC countries of Afghanistan, DRC, Malawi, Haiti, and Tanzania (between 2013 and 2018). The main objective of the study was to examine the overall satisfaction of caretakers of sick children during their interactions with healthcare providers at hospitals, primary care facilities, and health posts or dispensaries. The explanatory variables were characteristics of children and caretakers, health system foundations (health facility and healthcare provider characteristics), and process of care (clinical quality of care and perceived quality of care or user experience). The authors used multivariate logistic regression analyses and found that user experience explained a larger proportion of variance in overall satisfaction with care provided from health facilities, with considerable differences between countries. They reported that likelihood of being very satisfied (overall) with services was negatively affected by issues caretakers had with cost of services, availability of medications, amount of explanation provided by healthcare provider, waiting time, cleanliness of health facility, and how healthcare provider treated them. Clinical quality of care was a significant predictor of overall user satisfaction with services offered. The authors also provided countryspecific results, as supplementary material. A comparison was made in findings (overall) between countries by including a categorical variable for countries (Malawi as the ref group). This is an important study with its findings having the potential to influence health policies and
--

	interventions to improve quality of care, and patient and user experience, as well as patient satisfaction with health services and enhance utilization of maternal and child healthcare in LMIC settings. However, before the paper is considered for publication, I have encountered major issues that the authors should address them in order to make the paper readable for an average audience, especially health managers and policy makers. A major issue I have encountered is the statistical analysis that the authors focused too much on statistical terminology and formulae, while the authors could provide one statistical model, based on their theoretical framework, as is in page 9 (line 20), and avoid providing formulae on variance calculation, R2, and intraclass correction (page 10). Even though, the model in page 9 hasn't been properly specified (see my comments in statistical analysis). The authors should also work more on the discussion (see my comments in the discussion). Abstract Objective: The authors may opt to be more specific that they are interested in assessing variations in characteristics related to health facility and healthcare professional, clinical quality of care, user experience, and sociodemographic of children and their caretakers in relation to overall user satisfaction with quality of care they received from health facilities and hospitals. Age of children should also be specified from the beginning (e.g., < 5 years) Line17 "... pooled cross-sectional" Line 21 "... very satisfied vs. not very satisfied or dissatisfied." Because the responses were collected on a 3-level Likert scale question Results: Description of the patients and caretakers, health facilities and healthcare providers is missing. Important findings on user experience with cost of services, availability of medicines, amount of explanation provided by healthcare provider, waiting time, cleanliness of health facility, and how healthcare provider treated the patients and caretakers are missing. Country level analyses should be reported on major findings here. Discussion: It has focused on user experience, which needs to be supported by the findings reported in previous paragraph. Also, throughout the paper, the authors should use either user satisfaction or caretaker satisfaction with health services. It needs to describe from the beginning that overall satisfaction was measured and used as the outcome, because previous studied in Afghanistan or some other countries examined patient satisfaction with various (specific) aspects of perceived quality of care. Introduction The sentence "User satisfaction is the heart of high-quality health system." maybe a strong statement,
--	---

	as not all researchers involved in patient satisfaction and quality of care agree with it. There are researchers who question this statement and think that technical or clinical quality of care is more important than user or patient satisfaction. In the opening of the second paragraph, the authors seem to contradict what they stated in the first paragraph. Because the authors questioned whether user satisfaction is an appropriate measure as it is influenced by health system related factors, which include technical quality of care. Lines 25-26 “single-level analyses and therefore was unable to determine how much of the variation in user satisfaction was explained at the facility level versus the consultation level” may not be true, because clustering of information related to consultation occurs at the facility level (e.g., in a hospital where there are several doctors, nurses, midwives, and allied health workers, the quality of consultation might be similar between the doctors, between nurses, and between midwives); therefore, results from the multivariate analyses can be adjusted for the intraclass correction at the health facility level. Thus, a single-level analysis where clustering effects have been adjusted can be valid and the statistical approach used can be appropriate method. Methods Line16/page6: Age “for sick children” needs to be specified. Framework: if it can be depicted in a figure, it can be more readable. Line52/page7: Age of child needs to be specified (e.g., <=12month, 13-xxmonths, age not specified). Line6/page8: caretaker’s age (0-19) seems strange. Is it x-19years or <19years? Line55/page8: the country level variables need to be described (e.g., how they were created) Statistical analysis: The main issue I have encountered is the use of a two-level logistic regression, and the specification of the statistical equation (lines 19-20/page9). As I stated earlier, cluster effects of data pertaining to quality of consultation can be addressed by the inclusion of a random cluster effect in the logistic regression at the health facility level (because consultations take place within health facilities). The standard errors and related statistics will be adjusted once the random cluster effects added to the multivariate logistic regression model (the authors may have done it already). It seems that the statistical model (lines 19-20) doesn’t have the proper notations. For example, β_1 refers to the coefficient from a predictor variable of X which is among the X_{ij}. What about the coefficient from a second, third, fourth, and so on Xs? Also I am not convinced with the use of the term “Random-effects parameter U_{0j}”; Why it is U_0 ? Shouldn’t it be U_{ij} ”? because it refers to the unobservable factors for observation i that affects Y_{ji}. The
--	--

	authors can have a look at the Textbook of Introductory Econometrics by Jeffery M Wooldridge. The equations in page10 confuse the reader, and the authors may opt to use the equation on variance (lines 15-16). Results I wonder why there is Table1 which isn't based on the findings from this study? If the authors want, they can present some of the information from this table in the introduction, but not as part of the results. Line31 "We performed a complete case analysis" is not relevant here. The sentence should be part of the method. The results should start from description of characteristics of patients, caretakers, health facilities, healthcare providers, process of care (patient experience, clinical quality), and caretaker satisfaction (its distribution can be shown in graphs). The term "analytical" before the word "sample" is confusing; just saying "sample" is enough. Also, the term "caretaker" should be used consistently. In line 8/page14, it says "caregiver". Page16: Do the authors need to present or report variations on residual intraclass correction? Page17: To me, this page presents the main findings of this study. The authors may even think about presenting the findings in a forest plot graph in order to make them more readable for an avg reader. Lines 4-14/page20: the results are not reported here; the main findings should be reported here. Discussion The authors can start the first paragraph as a reflection of the findings in this study (it is missing now). The paragraph "To our knowledge, this is the first multi-country ..." can be the second paragraph. Lines 20-21/page22 "The influence of cultural differences on patient satisfaction needs to be explored in future research" can be moved to the end of the discussion as part of a separate paragraph on "future research". The authors can briefly discuss the bias potentially occurring during patient – provider interaction while the provider is being observed for their performance. Also, it should be described that caretakers who were "more or less satisfied" were coded as "not satisfied", despite consideration the social desirability bias potentially occurring during the exit interviews. Conclusion It should start from a sentence related to caretaker experience and its effects on caretaker satisfaction with quality of care at health facilities and hospitals in light of the findings in this study. Then, the authors can bring statements related to the positive impact of user experience and patient satisfaction on the health system and utilization of maternal and child healthcare, particularly in LMIC settings. Lines 13-14/page23 can be moved to the paragraph on "future research" at the end of discussion.
--	--

REVIEWER	Nigusso, Fikadu Tadesse
-----------------	-------------------------

	World Food Programme, Programme
REVIEW RETURNED	20-Feb-2023

GENERAL COMMENTS	Review comments: The manuscript is well-discussed. I recommend the manuscript be accepted with minor revisions.  In this manuscript, the authors attempted to study an important health topic, the proportion of variance in user satisfaction as explained by the health system and non-health system factors in low- and middle-income countries (LMICs) remains unclear. There are a few remarks that should be considered to improve the quality of the manuscript. Overall, the method section has to be revised and rewritten. 1. Background  As user satisfaction levels and context varies countries, most evidence presented in the background is not from low- and middle-income countries. Is this due to a lack of data or another reason? 2. Methods  The study was described to be available at the Service Provision Assessment (SPA) surveys conducted by the Demographic and Health Survey (DHS) Program of the United States Agency for International Development (USAID) as referred to available at: https://dhsprogram.com/Methodology/SurveyTypes/SPA.cfm. But, to crosscheck this study with country-specific survey data, the survey data is not available on the server for countries such as DRC at the time of this review. Also, it was stated that "...We limited our study population to the most recent SPA surveys that included a question on user satisfaction with consultation for sick children in five countries". However, a 2013 DHS data is used for Malawi and Tanzania while the 2017 and 2015 data are available that nullifies the above statement on the use of the most recent SPA. Therefore, consider revising your data. The method itself is not clear and not painstakingly thorough and incorporated the use of sufficient descriptions. For instance, it was described that "the facilities are selected from a comprehensive list of health facilities in each country, categorized by facility type, managing authority, and region while a nationally representative sample of health facilities is selected." This is a multilevel systematic review where study participants and health facilities are not directly selected but secondary data/survey. As such, discussing the details of how the facilities were selected seems less important. Likewise,
---

	it is stated that "...A sample of health providers was selected from the facility roster to be interviewed and observed. Trained enumerators observed clinical visits to assess adherence to clinical guidelines during consultations for sick children". To my understanding, this study employed secondary data from nationally conducted DHS. Thus, how are the sampling strategy and data management method employed in a such multilevel analysis?  • It was described that the data are openly available, however, data quality management issues were not discussed. Data validity and reliability safeguarding methods employed were not described. Therefore, it's recommendable to address how data quality was handled. • What are the precise inclusion and exclusion criteria of the study? 3. Result  • In this multilevel factorial analysis, the four key assumptions of multivariate analysis: independence, linearity, normality, and homoscedasticity were not discussed in the result section. Therefore, it's recommended to recheck this section. Also, rechecking the statistical results and figures is recommended.
--	--

VERSION 1 – AUTHOR RESPONSE

Reviewer 1 comment: The statistical analysis focused too much on statistical terminology and formulae, while the authors could provide one statistical model, based on their theoretical framework, as is in page 9 (line 20), and avoid providing formulae on variance calculation, R², and intraclass correlation (page 10). Even though, the model in page 9 hasn't been properly specified (see my comments in statistical analysis).

Authors' response: We thank the reviewer for this comment. We simplified the statistical terminology and formulae. We revised the statistical model based on our theoretical framework. We moved the formula on variance calculation and intraclass correlation to the appendix for readers who want to know more or who would be interested in replicating this study.

Reviewer 1's comment: Abstract Objective: The authors may opt to be more specific that they are interested in assessing variations in characteristics related to health facility and healthcare professional, clinical quality of care, user experience, and sociodemographic of children and their caretakers in relation to overall user satisfaction with quality of care they received from health facilities and hospitals.

Authors' response: We followed BMJ Open's guidelines which request a 300-word structured abstract. We reached 300 words just by covering the essential information needed in an abstract. Thus, we are not able to include more details without going over the word limit. However, the abstract does state that: "Predictors pertained to child and caretaker characteristics, health system foundations, and process of care (e.g., care competence, user experience)." Moreover, the additional information requested by the reviewer is well described in the introduction, methods, and results of the manuscript.

Reviewer 1's comment: Age of children should also be specified from the beginning (e.g., < 5 years)

Authors' response: We modified the abstract to specify the age group from the beginning.

Reviewer 1's comment: Line17 "... pooled cross-sectional"

Authors' response: As suggested, we added the word "pooled" before the name of the cross-sectional design.

Reviewer 1's comment: Line 21 "... very satisfied vs. not very satisfied or dissatisfied." Because the responses were collected on a 3-level Likert scale question.

Authors' response: As requested, we modified the text to indicate that the outcome variable was whether the child's caretaker was very satisfied versus more or less satisfied or not satisfied.

Reviewer 1's comment: Abstract results: Description of the patients and caretakers, health facilities and healthcare providers is missing.

Authors' response: BMJ Open requires a 300-word structured abstract which is quite condensed. Thus, we are only able to present the results that are directly related to the study's primary objectives. Any additional information would make us go over the word limit. However, please be assured that detailed descriptions of the patients, caretakers, health facilities and healthcare providers are provided in the results section of the manuscript.

Reviewer 1's comment: Abstract results: Important findings on user experience with cost of services, availability of medicines, amount of explanation provided by healthcare provider, waiting time, cleanliness of health facility, and how healthcare provider treated the patients and caretakers are missing. Country level analyses should be reported on major findings here.

Authors' response: We modified the abstract to include important findings on user experience, including cost of services, availability of medicines and amount of explanation provided by healthcare provider. However, due to the 300-word limit required by *BMJ Open*, we were not able to add a lot of specific findings from the country level analyses. We do highlight that "Country level analyses showed differences in variance explained and in associations with predictors" in order to invite the reader to read the article.

Reviewer 1's comment: Abstract discussion: It has focused on user experience, which needs to be supported by the findings reported in previous paragraph.

Authors' response: We modified this section of the abstract to make it more aligned and supported by the findings reported in the results section.

Reviewer 1's comment: Also, throughout the paper, the authors should use either user satisfaction or caretaker satisfaction with health services. It needs to describe from the beginning that overall satisfaction was measured and used as the outcome, because previous studied in Afghanistan or some other countries examined patient satisfaction with various (specific) aspects of perceived quality of care.

Authors' response: We modified the abstract and the study's objectives in the manuscript to specify that we are referring to overall caretaker satisfaction.

Introduction

Reviewer 1's comment: The sentence "User satisfaction is the heart of high-quality health system." maybe a strong statement, as not all researchers involved in patient satisfaction and quality of care agree with it. There are researchers who question this statement and think that technical or clinical quality of care is more important than user or patient satisfaction. In the opening of the second paragraph, the authors seem to contradict what they stated in the first paragraph. Because the authors questioned whether user satisfaction is an appropriate measure as it is influenced by health system related factors, which include technical quality of care.

Authors' response: We toned-down this sentence by simply stating that user satisfaction is an important dimension of high-quality health system. We also added a reference to the Lancet Global Health Commission on High Quality Health Systems which supports the statement.

Reviewer 1's comment: Lines 25-26 "single-level analyses and therefore was unable to determine how much of the variation in user satisfaction was explained at the facility level versus the consultation level" may not be true, because clustering of information related to consultation occurs at the facility level (e.g., in a hospital where there are several doctors, nurses, midwives, and allied health workers, the quality of consultation might be similar between the doctors, between nurses, and between midwives); therefore, results from the multivariate analyses can be adjusted for the intraclass correction at the health facility level. Thus, a single-level analysis where clustering effects have been adjusted can be valid and the statistical approach used can be appropriate method.

Authors' response: This sentence was removed from the introduction because it wasn't clear.

Reviewer 1's comment: Methods : Line16/page6: Age "for sick children" needs to be specified.

Authors' response: We modified the manuscript to specify the age of the children.

Reviewer 1's comment: Framework: if it can be depicted in a figure, it can be more readable.

Authors' response: We inserted Figure 1 in the manuscript to illustrate the framework and make it more readable.

Reviewer 1's comment: Line52/page7: Age of child needs to be specified (e.g., <=12month, 13-xxmonths, age not specified).

Authors' response: We clarified the age groups. The manuscript now states the following ages: 12 months or less, 13 months to 59 months, and age not specified.

Reviewer 1's comment: Line6/page8: caretaker's age (0-19) seems strange. Is it x-19years or <19years?

Authors' response: We clarified the age groups for the caretakers. The manuscript now states the following age groups: ≤19 years, 20–35 years, ≥ 36 years, and age not specified.

Reviewer 1's comment: Line55/page8: the country level variables need to be described (e.g., how they were created)

Authors' response: There was a typing error on that line. The word “variables” should have been singular. We corrected the text to this: “We included the country variable as a proxy for country-level outer context.”

Reviewer 1's comment: Statistical analysis: The main issue I have encountered is the use of a two-level logistic regression, and the specification of the statistical equation (lines 19-20/page9). As I stated earlier, cluster effects of data pertaining to quality of consultation can be addressed by the inclusion of a random cluster effect in the logistic regression at the health facility level (because consultations take place within health facilities). The standard errors and related statistics will be adjusted once the random cluster effects added to the multivariate logistic regression model (the authors may have done it already).

Authors' response: We thank the reviewer for the thought on adjusting the clustering effect. We are well aware that just correcting for clustering can also address the over-rejection arising from the problem that individuals within a cluster are more alike. However, some evidence indicates that multilevel modeling is a more effective estimation strategy than just adjusting for clustering effect (see reference #1 below). It was found that even after correcting for clustering, there is a tendency to over-reject the null hypothesis of no effect. Bertrand, Duflo and Mullainathan (2004) note that in order to correct for the standard error due to clustering, the presence of common random effect at the group level needs to be accounted for (see ref #2).

1. Gelman, Andrew and Jennifer Hill (2007), *Data Analysis Using Regression and Multilevel/Hierarchical Models*, Cambridge University Press.
2. Bertrand, Marianne, Esther Duflo and Sendhil Mullainathan (2004), *How Much Should We Trust Differences-in-Differences Estimates?*, *The Quarterly Journal of Economics* (Vol. 119(1)), pages 249-275, February.
3. Cheah, B. C., (2009). Clustering standard errors or Modeling Multilevel Data?

Reviewer 1's comment: It seems that the statistical model (lines 19-20) doesn't have the proper notations. For example, β_1 refers to the coefficient from a predictor variable of X which is among the X_{ij} . What about the coefficient from a second, third, fourth, and so on Xs?

Authors' response: We thank the reviewer for this opportunity to correct the notation. We fixed it so that all of the variables are represented.

Reviewer 1's comment: I am not convinced with the use of the term “Random-effects parameter U_{0j} ”; Why it is U_0 ? Shouldn't it be U_{ij} ”? because it refers to the unobservable factors for observation i that affects Y_{ij} . The authors can have a look at the Textbook of Introductory Econometrics by Jeffery M Wooldridge.

Authors' response: We thank the reviewer for taking the time to review the equations carefully. We verified and the term U_{0j} is correct. U_{0j} is the residual differential for facility j where 0 means a grand mean. If it was multilevel linear regression, we would include an individual-level residual in the specification like this:

$$y_{ij} = \beta_0 + \beta_{1i}x_{1ij} + (u_{0j} + e_{0ij})$$

However, level-1 residual term has been left out of the specification since logistic regression models do not have a level-1 residual term. The reviewer can refer to many papers that have used this specification (for linear multilevel model please see reference #1 below and for logistic multilevel model, see reference #2 and #3). More references and textbooks are also available online. For example, reference #4 is a textbook that uses the term U_{0j} .

1. Kim, Rockli, et al. "Contribution of socioeconomic factors to the variation in body-mass index in 58 low-income and middle-income countries: an econometric analysis of multilevel data." *The Lancet Global Health* 6.7 (2018): e777-e786.
2. Kim, Rockli, Sanjay K. Mohanty, and S. V. Subramanian. "Multilevel geographies of poverty in India." *World Development* 87 (2016): 349-359.
3. Jain, Anoop, et al. "Small area variations in dietary diversity among children in India: a multilevel analysis of 6–23-month-old children." *Frontiers in Nutrition* 8 (2022): 1353
4. Anderson, D., (2012). Hierarchical Linear Modeling (HLM): An Introduction to Key Concepts Within Cross-Sectional and Growth Modeling Frameworks. Behavioral Research and Teaching.

Reviewer 1's comment: The equations in page10 confuse the reader, and the authors may opt to use the equation on variance (lines 15-16).

Authors' response: We have moved these equations to the supplementary materials.

Results

Reviewer 1's comment: I wonder why there is Table1 which isn't based on the findings from this study? If the authors want, they can present some of the information from this table in the introduction, but not as part of the results.

Authors' response: We created a subsection in the methods called "study setting" and moved Table 1 in this section to avoid creating confusion between the findings from this study and the description of the context.

Reviewer 1 comment: Line31 "We performed a complete case analysis" is not relevant here. The sentence should be part of the method. The results should start from description of characteristics of patients, caretakers, health facilities, healthcare providers, process of care (patient experience, clinical quality), and caretaker satisfaction (its distribution can be shown in graphs).

Authors' response: We moved the section about complete case analysis to the beginning of the section called "Statistical analysis".

Reviewer 1's comment: The term "analytical" before the word "sample" is confusing; just saying "sample" is enough.

Authors' Response: We removed the word analytical.

Reviewer 1's comment: The term "caretaker" should be used consistently. In line 8/page14, it says "caregiver".

Authors' response: We reviewed the manuscript to replace the word "caregiver" with "caretaker".

Reviewer 1's comment: Page16: Do the authors need to present or report variations on residual intraclass correlation?

Authors' response: We agreed to delete the section on residual intraclass correlation as it was not essential.

Reviewer 1's comment: Page17: To me, this page presents the main findings of this study. The authors may even think about presenting the findings in a forest plot graph in order to make them more readable for an avg reader.

Authors' response : We thank the reviewer for this suggestion and replaced the table by a forest plot graph. However, please note that based on the study's objectives, our main finding pertains to "variation" of satisfaction, rather than the determinants of satisfaction.

Reviewer 1's comment: Lines 4-14/page20: the results are not reported here; the main findings should be reported here.

Authors' response: We modified the text to present the findings for each country and inserted Table 3 in the main manuscript.

Reviewer 1's comment: Discussion. The authors can start the first paragraph as a reflection of the findings in this study (it is missing now). The paragraph "To our knowledge, this is the first multi-country ..." can be the second paragraph.

Authors' response: We modified the beginning of the discussion by adding the following paragraph:

"Based on combined data from five countries, we found that factors related to the process of care during consultations for children under the age of five explained the highest proportion of variance in caretaker satisfaction (13.8 %), followed by factors related to the health systems' foundations (3.8%). Patient and caretakers' characteristics explained the smallest proportion of variance in caretaker satisfaction (0.9%). The children's number of symptoms and the caregiver's higher education levels both decreased the odds of being very satisfaction. The facility type and the readiness to provide sick child services also predicted overall satisfaction, although the providers' characteristics did not. In terms of process of care, the odds of being very satisfied were lower for caretakers who were not given adequate explanations and those who had encountered problems with medication availability and the cost of services."

Reviewer 1's comment: Lines 20-21/page22 "The influence of cultural differences on patient satisfaction needs to be explored in future research" can be moved to the end of the discussion as part of a separate paragraph on "future research". Lines 13-14/page23 can be moved to the paragraph on "future research" at the end of discussion.

Authors' response: We added a paragraph on future research at the end of the discussion and moved the two sentences as requested. This paragraph states the following:

"The study is useful to develop a research agenda. First, in our analyses, there is still a substantial proportion of unexplained variation in caretakers' satisfaction. Future research should explore new determinants of variation in patient satisfaction throughout children's care pathways. Second, the heterogeneity between countries supports the need for more studies on how cultural differences affect patient satisfaction."

Reviewer 1's comment: The authors can briefly discuss the bias potentially occurring during patient – provider interaction while the provider is being observed for their performance.

Authors' response: We added a sentence to make this bias more explicit. The paragraph now starts like this: *"This study has several limitations. First, the satisfaction rates reported by caretakers were generally high, possibly due to desirability bias in participants' responses during interviews, low patient expectations, and behaviors during the audits. The observation of the providers' performance during the consultation potentially biased the patient – provider interaction."*

Reviewer 1's comment: Also, it should be described that caretakers who were "more or less satisfied" were coded as "not satisfied", despite consideration the social desirability bias potentially occurring during the exit interviews.

Authors' response: We modified the methods section to clearly explain the following: "Based on the low levels of dissatisfaction, we created a binary variable of satisfaction equal to 1 if the caretaker was very satisfied and 0 otherwise (i.e., if the caretaker was more or less satisfied, or not satisfied)."

Reviewer 1's comment: Conclusion. It should start from a sentence related to caretaker experience and its effects on caretaker satisfaction with quality of care at health facilities and hospitals in light of the findings in this study. Then, the authors can bring statements related to the positive impact of user experience and patient satisfaction on the health system and utilization of maternal and child healthcare, particularly in LMIC settings.

Authors' response: As requested, we modified the conclusion to add a sentence related to caretaker experience and its effects on caretaker satisfaction. The conclusion starts with the following : "This study showed that the process of care explained the largest proportion of variance in caretaker satisfaction during sick child consultations in five LMICs. Caretakers who encountered problems related

to waiting time, how the staff treated them, the amount of explanation provided, medication availability and cost of services were less likely to be very satisfied.”

Reviewer 2

Reviewer 2's comment: 1. Background. As user satisfaction levels and context varies countries, most evidence presented in the background is not from low- and middle-income countries. Is this due to a lack of data or another reason?

Authors' response: We added a sentence to explain that most evidence on the determinants of variation in caretaker satisfaction with sick child consultations comes from high-income countries (see page 4). This is specifically the research gap that we are trying to fill by using data from five LMICs. However, the literature that we referenced from high-income countries is still useful to understand the state of knowledge and to make comparisons.

Reviewer 2's comment: The study was described to be available at the Service Provision Assessment (SPA) surveys conducted by the Demographic and Health Survey (DHS) Program of the United States Agency for International Development (USAID) as referred to available at: <https://dhsprogram.com/Methodology/SurveyTypes/SPA.cfm>. But, to crosscheck this study with countryspecific survey data, the survey data is not available on the server for countries such as DRC at the time of this review.

Authors' response: We modified the data availability statement to facilitate access to the original database. The statement now states the following: The Service provider data used for these analyses are publicly available through the Demographic and Health Survey (DHS) Program of United States Agency for International Development (USAID). Before you can download datasets, you must register as a DHS data user on this website: <https://dhsprogram.com/data/new-user-registration.cfm>. We added references to each specific dataset, following the journal's preferred format, and mentioned that the data for each country can be accessed via the following websites:

Afghanistan: https://dhsprogram.com/data/dataset/Afghanistan_SPA_2018.cfm

Democratic Republic of the Congo (DRC): https://dhsprogram.com/data/dataset/Congo-Democratic-Republic_SPA_2017.cfm

Haiti: https://dhsprogram.com/data/dataset/Haiti_SPA_2017.cfm

Malawi: https://dhsprogram.com/data/dataset/Malawi_SPA_2013.cfm

Tanzania: https://dhsprogram.com/data/dataset/Tanzania_SPA_2014.cfm

Reviewer comment: Also, it was stated that "...We limited our study population to the most recent SPA surveys that included a question on user satisfaction with consultation for sick children in five countries". However, a 2013 DHS data is used for Malawi and Tanzania while the 2017 and 2015 data are available that nullifies the above statement on the use of the most recent SPA. Therefore, consider revising your data.

Authors' response: The DHS data from Malawi and Tanzania that the reviewer mentioned is not actually from the Service Provider Assessment. It is data from other types of surveys (e.g., MIS, standard DHS) that are not related to this study. The manuscript is correct. We did use the most recent Service provider assessment data available from Malawi and Tanzania. The information on the types of surveys and the years when they were conducted is available here:

<https://dhsprogram.com/Methodology/survey-search.cfm?pgtype=main&SrvyTp=country>

Reviewer 2's comment: The method itself is not clear and not painstakingly thorough and incorporated the use of sufficient descriptions. For instance, it was described that "*the facilities are selected from a comprehensive list of health facilities in each country, categorized by facility type, managing authority, and region while a nationally representative sample of health facilities is selected.*" This is a multilevel systematic review where study participants and health facilities are not directly selected but secondary data/survey. As such, discussing the details of how the facilities were selected seems less important. Likewise, it is stated that "...A sample of health providers was selected from the facility roster to be interviewed and observed. Trained enumerators observed clinical visits to assess adherence to clinical guidelines during consultations for sick children". To my understanding, this study employed secondary data from nationally conducted DHS. Thus, how are the sampling strategy and data management method employed in a such multilevel analysis? It was described that the data are openly available, however, data quality management issues were not discussed. Data validity and reliability safeguarding

methods employed were not described. Therefore, it's recommendable to address how data quality was handled.

Authors' response: The reviewer is correct in saying that we did not directly collect the data. We employed secondary data from nationally conducted surveys. We added the following paragraph in the methods section regarding data management and quality control and included references to each Service Provision Assessment Survey :

"The procedures for data management and quality were similar across countries [6–10]. The interviewers were supposed to review the data and enter it in tablets. The data files were transferred to a supervisor who oversaw the data collection process. When supervisors noted missing information or errors, they sent the data back to the interviewer for revision. Then, the data was sent to a central office via the Internet. In the central office, data processors detected inconsistencies and gave feedback to the team in the field to resolve the problems. For tracking of systematic errors arising from each interviewer, field check tables were run. If an interviewer committed errors systematically, the central office and coordinators followed up with the interviewers [6–10]."

Reviewer 2's comment: What are the precise inclusion and exclusion criteria of the study?

Authors' response: In the methods section, we added the following text to specify the inclusion and exclusion criteria:

"All caretakers who sought care for a sick child consultation in hospitals or primary health care facilities were included in our study without any specific exclusion criteria."

Reviewer 2's comment: Result. In this multilevel factorial analysis, the four key assumptions of multivariate analysis: independence, linearity, normality, and homoscedasticity were not discussed in the result section. Therefore, it's recommended to recheck this section.

Authors' response: We did not conduct a multilevel factorial analysis, but rather a multilevel logistic regression. Logistic regression does not require a linear relationship between the dependent and independent variable. However, there should be a linear relationship between any continuous independent variable and the logit transformation (aka log-odds) of the dependent variable. We conducted a Box-Tidwell test to examine this relationship. We conducted a Box-Tidwell test and added a sentence in the methods section stating that *"We confirmed the linearity between the four continuous independent variables and log-odds of the outcome using the Box-Tidwell test."*

Homoscedasticity and normality of residual is required in linear regression but not for logistic regression.

Reviewer 2's comment: Also, rechecking the statistical results and figures is recommended.

Authors' response: The first author and corresponding author rechecked the statistical results and figures.

VERSION 2 – REVIEW

REVIEWER	Stanikzai, Muhammad Haroon Kandahar University, Public Health
REVIEW RETURNED	20-Oct-2023
GENERAL COMMENTS	I would like to thank you for your efforts.